# Feasibility of Heart Rate and Respiratory Rate Estimation by Inertial Sensors Embedded in a Virtual Reality Headset

**DOI:** 10.3390/s20247168

**Published:** 2020-12-14

**Authors:** Claudia Floris, Sarah Solbiati, Federica Landreani, Gianfranco Damato, Bruno Lenzi, Valentino Megale, Enrico Gianluca Caiani

**Affiliations:** 1Electronics, Information and Bioengineering Department, Politecnico di Milano, 20133 Milano, Italy; claudia.floris@polimi.it (C.F.); sarah.solbiati@polimi.it (S.S.); federica.landreani@polimi.it (F.L.); 2Softcare Studios Srls, 00137 Rome, Italy; g.damato@softcarestudios.com (G.D.); b.lenzi@softcarestudios.com (B.L.); v.megale@softcarestudios.com (V.M.); 3Consiglio Nazionale delle Ricerche, Istituto di Elettronica e di Ingegneria dell’Informazione e delle Telecomunicazioni, 20133 Milano, Italy

**Keywords:** heart rate, respiratory rate, virtual reality headsets, gyroscope, accelerometer, ballistocardiography

## Abstract

Virtual reality (VR) headsets, with embedded micro-electromechanical systems, have the potential to assess the mechanical heart’s functionality and respiratory activity in a non-intrusive way and without additional sensors by utilizing the ballistocardiographic principle. To test the feasibility of this approach for opportunistic physiological monitoring, thirty healthy volunteers were studied at rest in different body postures (sitting (SIT), standing (STAND) and supine (SUP)) while accelerometric and gyroscope data were recorded for 30 s using a VR headset (Oculus Go, Oculus, Microsoft, USA) simultaneously with a 1-lead electrocardiogram (ECG) signal for mean heart rate (HR) estimation. In addition, longer VR acquisitions (50 s) were performed under controlled breathing in the same three postures to estimate the respiratory rate (RESP). Three frequency-based methods were evaluated to extract from the power spectral density the corresponding frequency. By the obtained results, the gyroscope outperformed the accelerometer in terms of accuracy with the gold standard. As regards HR estimation, the best results were obtained in SIT, with R_s_^2^ (95% confidence interval) = 0.91 (0.81−0.96) and bias (95% Limits of Agreement) −1.6 (5.4) bpm, followed by STAND, with R_s_^2^ = 0.81 (0.64−0.91) and −1.7 (11.6) bpm, and SUP, with R_s_^2^ = 0.44 (0.15−0.68) and 0.2 (19.4) bpm. For RESP rate estimation, SUP showed the best feasibility (98%) to obtain a reliable value from each gyroscope axis, leading to the identification of the transversal direction as the one containing the largest breathing information. These results provided evidence of the feasibility of the proposed approach with a degree of performance and feasibility dependent on the posture of the subject, under the conditions of keeping the head still, setting the grounds for future studies in real-world applications of HR and RESP rate measurement through VR headsets.

## 1. Introduction

Virtual reality (VR) and augmented reality headsets represent state-of-the-art technologically advanced systems able to simulate real-word interactive experience through a combination of technologies [1]. In particular, in the last 25 years, VR technologies have been expanded to cover a vast range of applications [2] with a widespread use also in the healthcare sector [3], ranging from physical rehabilitation [4] to psychiatric treatment for anxiety disorders (such as specific phobias and post-traumatic stress disorder [5]), pain management as a means to attenuate pain perception [6], anxiety and general stress during painful medical procedures (i.e., chemotherapy, dental and routine medical procedures) [7,8,9,10], disaster medicine [11], patient education [12] or for training purposes of healthcare workers [13,14].

In order to validate VR applications in a clinical setting, it is important to assess their impact in terms of quality of experience (QoE) defined as “the degree of delight or annoyance of the user of an application or service” [15]. Thus, it is related to the quality of rendering of an immersive multimedia in terms of impact on users’ emotion, sense of presence, level of induced stress and degree of engagement [16,17,18], which can eventually be used as feedback to provide the patient with a more personalized media experience. The way the user perceive the VR environment can be assessed explicitly or implicitly [19]: the former refers to QoE evaluation via post-test questionnaires [20] or pre-defined rating scales [21], thus resulting in a subjective metric of the user immersion level or of the VR-induced stress. The latter is a bio-inspired approach based on the acquisition of physiological signals from the VR user, such as electroencephalography (EEG), heart rate (HR) and respiratory rate, facilitating real-time monitoring of QoE without subjective biases [19]. In particular, there have been efforts to measure brain activity in order to understand QoE by using various types of EEG headsets [19], but their level of intrusiveness turned out to have a bad impact on the user’s QoE. Other approaches to capture biometrics while using VR have utilized non-invasive wearable devices. For instance, Egan et al. proposed the use of two consumer devices, Fitbit heart rate monitor and PIP Biosensor, to capture HR and electro-dermal activity to determine user emotional arousal while undergoing the immersive experience wearing a head mounted display Oculus Rift [22].

As a further advance in this field of research, the feasibility to directly extract cardiac and respiratory information from low-cost motion sensors (accelerometer, ACC, and gyroscope, GYR) already embedded in the Google Glass (Google, Inc.) head-worn device, a wearable, voice- and motion-controlled Android device that resembles a pair of eyeglasses, has been previously explored, showing promising results for non-intrusive physiological measurements by exploiting the ballistocardiographic method [23]. In fact, at each cardiac cycle, the force of blood acceleration, resulting from the flow into the ascending aorta and the carotids, generates a circular head movement as a reaction. These subtle motions associated with cardiac activity are imperceptible to the human eye but not to the sensor resolution of current micro-electromechanical systems technology embedded in wearable devices. When positioned at specific body locations, ACC and GYR sensors have shown their potential to capture peripheral motion associated with cardiac activity [24], also when embedded in common smartphone devices [25,26], and even being able to capture changes in sympathovagal balance induced by an external stressor [27], thus representing a complementary solution to assess the mechanical cardiac function and respiratory activity in a non-intrusive and electrodes-free way [28], while the subject is remaining relatively “still” [29].

The main drawback of such approaches is that the signal component relevant to the cardiac or respiratory activity is limited in amplitude, and can be easily occluded by noise and artifacts generated by body movements, thus highly restricting their applicability outside laboratory settings unless limiting to few opportunistic measurements in real-life environment [24]. However, in a context of a real VR experience also involving body movements, short time intervals where the subject’s head is induced to stay still (for example, by looking a fixed point on the display as a requirement to advance level) could be induced by careful design, thus leading to the opportunity to monitor cardiac and respiratory activity at certain epochs, following or preceding certain stimuli embedded into the VR experience.

Based on these considerations, we hypothesized that ACC and GYR sensors embedded in current VR headset technology could allow the extraction of reliable physiologic parameters that could be used to monitor the QoE perceived by the users while experiencing a VR environment, without the need of additional instrumentation.

Accordingly, our aim was to test the feasibility of deriving HR and respiratory (RESP) rate from tri-axial GYR and ACC sensors embedded in a head-worn VR device while wearing it in three different body postures (standing up, sitting on a chair and lying down in supine position) in order to investigate the performance compared to relevant gold standards (ECG and imposed breathing frequency, respectively) and to verify how each body posture might limit the ability to detect the physiological information of interest.

## 2. Materials and Methods

### 2.1. Study Population

Thirty healthy volunteers (18 females and 12 males, mean age 24 ± 2 years old, mean stature 169 ± 11 cm, mean body mass (64 ± 15 kg) were recruited to participate in this study. All subjects provided voluntary written, informed consent to the experimental protocol, previously approved by the Ethical Committee of the Politecnico di Milano (n. 17/2019, 10 September 2019) in accordance with the Declaration of Helsinki of 1975, as revised in 2000.

### 2.2. Data Collection

Each volunteer was tested in resting condition, considering three different body postures (sitting (SIT), standing up (STAND) and lying down supine (SUP)) while wearing a head-worn VR device (Oculus Go, Oculus, Microsoft, USA) (Figure 1) equipped with 2560 × 1440 5.5″ (538 ppi) fast-switching LCD screen with standard 60 Hz refresh and Quad-core Qualcomm Snapdragon 821 CPU (two 2.3 GHz Kryo HP cores and two 2.15 GHz Kryo cores). Oculus Go offers the possibility to acquire linear accelerations (m/s^2^) and angular rotations (rad/s) of the head along the lateral (X), longitudinal (Y) and transverse (Z) directions through a 3-axes ACC and a 3-axes GYR, respectively. The reference axes orientation of the embedded user tracking system is shown in Figure 1.

Each participant was first instructed on how to wear the VR headset and set subjectively its level of tightness. Once worn, he/she was asked to confirm the fit and comfort of the headset; then, before the beginning of acquisition, the participant was asked to perform a fast head rotation, thus verifying the absence of relative motion between the VR and the head.

The VR signals (ACC and GYR) acquisition was performed by an “ad-hoc” application .apk, developed by Softcare Studios Srl (Rome, Italy) using Unity 2019.1.4 (Editor/Runtime) and Oculus Integration Package 1.36 (SDK), that accesses directly the motion information through the VR CPU. The time interval between consecutive acquired samples is dependent of concomitant accesses to the CPU from simultaneously running processes in the VR. As a consequence, the mean sampling rate within a VR acquisition was found variable around a mean value of 170 Hz, ranging between a minimum of 52.1 Hz and a maximum of 1000 Hz.

#### 2.2.1. Protocol for Heart Rate Evaluation

The experimental protocol for HR evaluation consisted in the simultaneous recordings for 30 s duration of the VR signals and of the 1-lead ECG (Coala Heart Monitor, Coala, Sweden), used as reference. The Coala device performs two subsequent acquisitions of 30 s each, one from the chest (Measurement 1, M1) by keeping the Coala on the left side of the thorax slightly above the heart (ECG-Chest in Figure 2) and one from the thumb (Measurement 2, M2) during which the subject is asked to position his/her thumbs on the electrodes and stay still (ECG-Thumb in Figure 2). At the conclusion of the acquisition, the Coala output consisted of the two 30 s ECG signals, and for each the mean HR, that was used as reference for comparison.

The synchronization between the two devices (VR headset and Coala) was achieved by asking the subject to perform a fast head rotation on one side at the beginning and end of the ECG acquisitions. In this way, the corresponding motion artifacts (indicated as SYNC in Figure 3a) were easily detectable from the recorded VR signals (Figure 3b), thus allowing the automatic extraction of the portion of ACC and GYR corresponding in time with the acquired gold standard ECG-measurements (M1-Chest and M2-Thumb). A schematic representation of the protocol for HR evaluation is shown in Figure 3a.

This procedure was repeated first while SIT, then while STAND, and finally while SUP resting, with a period of at least 3 min in-between to allow for HR accommodation to the new posture-related stationary condition.

#### 2.2.2. Protocol for Respiratory Rate Evaluation

The RESP rate protocol (Figure 4a) was performed for each subject once the HR protocol was completed. It consisted of acquiring the VR signals during a controlled breathing protocol including three different respiratory rates (7, 10.5 and 14 breaths per min) for a duration of 50 s each, imposed by simultaneously listening to a rhythmic audio used for meditation, which guides the user in the inspiration and expiration phases (https://www.youtube.com/watch?v=aXItOY0sLRY). The different respiratory rates were obtained from the original one of 7 breaths per minute simply by reproducing the audio at 1.5× and 2.0× speed, respectively. Moreover, in this case, two head rotations (SYNC in Figure 4a) were performed in order to introduce a visible motion artifact on the acquired VR signals at the beginning and at the end of each session for synchronization purposes (Figure 4b). This protocol was repeated with the subject first laying down in SUP position with the head on a rigid support, then while SIT, and finally while STAND, with a period of 3 min in-between each posture.

### 2.3. Data Processing

All the signal processing needed to extract the HR and RESP rate from the acquired signals collected by the VR device were performed offline using the Signal processing Toolbox in MATLAB R2019a (MathWorks, Natick, MA, USA).

#### 2.3.1. Pre-Processing

The portion of interest in the signals was automatically identified by detecting the start and end SYNC artifacts on the y-axis of the ACC (Figure 3b), where they resulted more uniquely identifiable than the corresponding GYR axis (Figure 3c), where additional artifacts were sometimes present during the acquisition. Then, a resampling operation at 170 Hz was performed for both ACC and GYR signals.

As the signal morphology is particularly sensitive to involuntary body motions, to avoid computing HR or RESP from portions of signal corrupted by motion artifacts, the proposed solution bases the identification of these artifacts on the signal envelope of each GYR component through the Hilbert transform. In particular, the noise detection is obtained by applying the function “isoutlier” to identify those samples whose value exceeds 3 Standard Deviations (SD) from the mean.

As concerns the 30-s acquisitions for HR estimation, in case motion artifacts were found to lie within the central 15-s portion of the selected signal, the relevant component was automatically excluded from further analysis, as it could negatively compromise the algorithm performance. Finally, for the longer VR-acquisition relevant to RESP estimation, an artifacts-free portion of the signal of at least 15 s duration was selected.

#### 2.3.2. HR Estimation

Given a specific VR acquisition, the processing of *x*-, *y*- and *z*-axis for M1-Chest (or M2-Thumb) was divided into the following steps, separately for the ACC and the GYR signals (Figure 5):(i)A digital band-pass (10–13 Hz) second order Butterworth IIR filter was applied to each component of the acquired signals [19].(ii)The magnitude of the total acceleration (separately for linear and angular components) was calculated as the square root of the sum of the squared components at each sample (Equation (1)).
(1)$$\bar{a}=\sqrt[{2}]{a_{x}^{2}+a_{y}^{2}+a_{z}^{2}}$$

(iii)An additional band-pass (0.75–2.5 Hz) second order Butterworth filter, thus limiting the detection of HR in the range between 45 and 150 beats per minute (bpm), was applied to the magnitude vector [19].(iv)To estimate the mean HR from the magnitude vector, two traditional approaches in the frequency domain were proposed: the Fast Fourier Transform (FFT) and the Short-Time Fourier Transform (STFT).(v)In the former case, the HR was computed multiplying the frequency corresponding to the highest spectral power amplitude $f_{HR}$ by 60 (bpm); in the latter case, the HR estimate was obtained as the median of the estimated frequencies corresponding to the highest spectral power amplitude computed by applying the Fourier transform in consecutive 10 s span time windows with 9 s of overlap.

**Figure 5 sensors-20-07168-f005:**
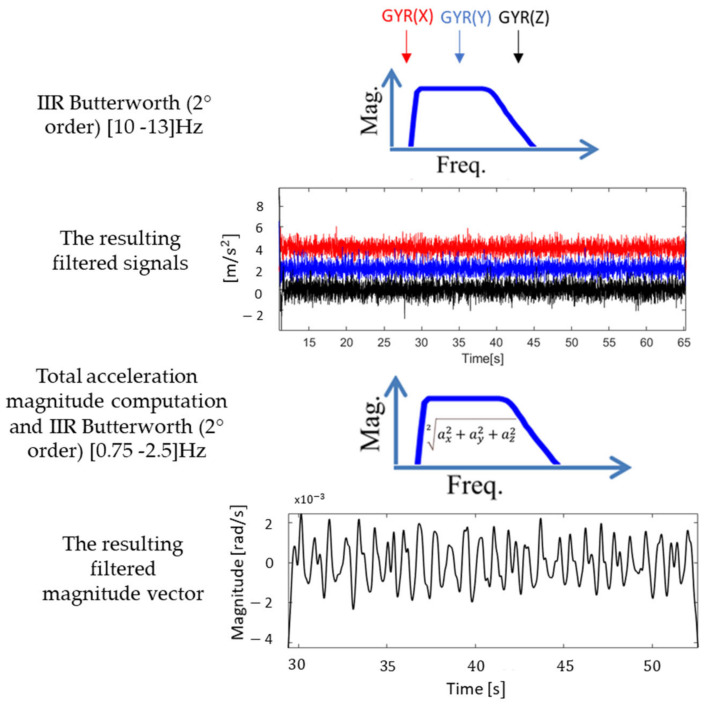
Example of the processing steps applied to the three GYR components (*x*-, *y*- and *z*-axis in red, blue and black, respectively) in a subject while sitting. After a first Butterworth filtering, the magnitude is computed, and then filtered again to result in the final signal.

Based on the consideration that the quasi-periodic characteristic of the cardiac-related head movement signals could lead to power spectral densities (PSDs) in which not only the fundamental frequency, but also its first harmonic could have a high power peak, a third modified step (Adjusted-FFT Method) was introduced in order to guide the automated selection of the correct peak on the power spectrum: in this case, the HR estimate was obtained by considering the ratio between the areas of the spectrum in the interval 0.75–1.5 Hz (Area1) and 1.5–2.25 Hz (Area2). For values of this ratio > 0.8 the algorithm hypothesizes the presence of a first harmonic with considerable power: it initially searches for the PSD maximum peak ($f_{HR}$) and, in case it falls in the frequency range >1.5 Hz (90 bpm), it searches for a local peak around $f_{HR}$/2 (Figure 6).

#### 2.3.3. Protocol for RESP Estimation

Regarding RESP estimation, the choice was to process only the GYR signal as movement artifacts were greatly affecting the ACC components. Given the longest artifacts-free (>15 s) portion of the GYR signal, identified within an average of 50 s of controlled-breathing acquisition, the three orthogonal components, which correspond to *x*-, *y*- and *z*-axis, were digitally filtered with a second order IIR Butterworth band-pass filter (0.1–0.9 Hz) in order to isolate the respiratory frequency component. Then, the frequency maximum peak, multiplied by 60 (breaths per minute (breaths/min)), was considered as the final RESP rate estimate for each axis (Figure 7).

### 2.4. Statistical Analysis

Due to the rejection of the null hypothesis of normality for HR results, to test the repeatability of the measurements, the mean HR obtained during the two 30-s periods in the same body posture were compared by Wilcoxon signed rank test, both for Coala (Chest vs Thumb) and for VR signals.

To determine the performance of the proposed method in terms of accuracy of the mean HR in the 30 s period, Spearman-rank order correlation analyses were performed between the corresponding measurements obtained by the VR and by the Coala, for both the ACC and the GYR signals, separately for M1-Chest and M2-Thumb acquisitions. To assess the agreement between measurements obtained with the two devices, Bland–Altman analysis was also performed, together with the computation of the standardized effect size.

In addition, the ACC and the GYR results were compared by Friedman Test, followed by post-hoc Wilcoxon signed rank test with Bonferroni correction, to evaluate potential postural-related differences.

To determine the performance of the proposed method in terms of accuracy of the RESP rate, the absolute difference between the estimated RESP and the imposed RESP rate was computed for each axis of the GYR signal, for data recorded while SUP, SIT and STAND. Then, to evaluate which axis contained more information relevant to the respiratory activity and to investigate potential postural related changes, the One-way ANOVA test was performed among the signal components for each imposed RESP rate.

All tests were considered statistically significant at *p* < 0.05. All statistical analyses were performed using the VassarStats program (http://vassarstats.net/).

## 3. Results

### 3.1. Results Relevant to Mean HR 

Due to the absence of the SYNC artifacts in the VR signals, thus preventing the identification of the portion to be analyzed, or to the non-proper skin contact by using the Coala ECG, resulting in absence of the gold standard reference, 4/30 subjects in SIT, 2/30 in STAND and 1/30 in SUP were discarded, thus resulting in a feasibility of our experimental protocol for HR evaluation equal to 87% in SIT, 93% in STAND and 97% in SUP.

Among the remaining subjects, the HR estimations were obtained from artifact-free signal portions with duration equal to 23 ± 2 s (Mean ± SD).

Considering the different body postures, in 25/30 subjects, the paired comparisons for all three postural conditions were possible. The Coala gold standard showed the expected changes in mean HR between SIT and STAND and between STAND and SUPINE (Figure 8), with a significant difference between the first 30 s (ECG Chest) and the following one (ECG Thumb) while in STAND (see complete results in Appendix A, Table A1). Accordingly, we decided to report separately for M1-Chest and M2-Thumb the results of the comparisons with the mean HR obtained from the VR signals.

As regards the ability of the VR signals in showing the expected changes in HR due to the different body postures, Figure 9 shows the cumulative results extracted from ACC (top panel) and GYR (bottom panel) in each posture, for each of the three implemented methods of analysis. It is possible to notice that, besides a trend of HR in STAND always higher than in SIT and SUP, only with the Adjusted-FFT method statistically significant results were obtained from the GYR signal, in line with what expected from the gold standard (see Appendix A, Table A2 for individual values).

Linear regression computed using Spearman rank-order correlation and Bland-Altman results are reported for ACC and GYR in Table 1 and Table 2, respectively, for each posture (SIT, STAND, SUP) and considering the results from the first (M1 Chest) and the second (M2 Thumb) acquisition separately.

The HR obtained from the ACC signal showed no or low correlation, higher bias and larger limits of agreement with the gold standard in each of the examined postures, and for each method, compared to the measurements obtained from the GYR, with the best performance obtained in SIT using the Adjusted FFT method, and the worst ones while in SUP, independently of the method used, while in STAND moderate correlation, no bias, and relatively narrow limits of agreements were found.

In Figure 10, the corresponding Bland–Altman analysis for the best performance (GYR signal in SIT position analyzed by Adjusted FFT method) is reported as example.

### 3.2. Results Relevant to RESP Rate

Several motion artifacts, dependent from the different postures, were observed in these GYR acquisitions that precluded the feasibility of measuring the respiratory rate in 2/90 acquisitions when in SUP, in 35/90 acquisitions while in SIT and 63/90 acquisitions while in STAND, thus resulting in a feasibility of 98%, 60% and 30%, respectively. In Table 3, the detail of the number of acquisitions that needed to be discarded is reported.

The results in terms of the median (25th; 75th) of the RESP rate estimated from each of the tri-axis components of the GYR signal, for each body posture, are reported in Table 4 for each imposed reference rate, set at 7, 10.5 and 14 breaths/min, while in Table 5 the median absolute error is reported, computed for each GYR component in the three postures.

It is possible to observe that the imposed frequency of 10.5 breaths/min was correctly identified in all postures by all GYR components, while some imprecisions were present for lower and faster rates. Moreover, the *x*-axis appeared the one with the lower estimation error for all postures and imposed frequencies.

## 4. Discussion

To the authors’ knowledge, this is the first study in which the feasibility of deriving mean HR and respiratory (RESP) rate from subtle head motion associated with the mechanical activity of the heart, measured by tri-axial gyroscope (GYR) and accelerometer (ACC) sensors embedded in a VR head-worn device in different postures, as an implicit measure of QoE, has been explored to set the groundwork for future studies in real-world applications.

Based on the obtained results, the main findings of this study are:

(1) In all postures, the ACC signal was outperformed by the GYR, which resulted in higher correlations, smaller biases and narrower confidence intervals when compared to gold standard HR obtained by the ECG, independently of the frequency analysis method applied.

(2) Considering the different postures analyzed, both for the gold standard and the corresponding estimated mean HR by GYR obtained by the Adjusted FFT-Method, the expected changes (i.e., HR SUP < SIT < STAND) elicited by the different orthostatic pressure gradient between the heart and the head, resulting in a corresponding autonomic system activation [31], were visible while non-significant differences were found using the FFT and STFT methods.

(3) The accuracy of GYR in estimating the HR varied according to the different posture and method of analysis, with the best results obtained while the subject was sitting using the Adjusted FFT method (LoA in a range between 5 and 7 bpm), while in standing a worsening of the performance was found (LoA between 12 and 15 bpm), with both the FFT and STFT methods performing similarly and better than the Adjusted-FFT one, probably due to the presence of additional noisy components in the PSD, relevant to body stabilization movements that were not adequately compensated. In supine, the worst performance for GYR was evidenced (LoA around 20 bpm), probably due to the dampening of the subtle head movements as the head was not anymore free to oscillate in 3D.

(4) For the RESP estimation, the ACC data were not suitable to extract breathing information, due to the presence of frequency components related to noise that prevented the identification of the respiratory activity peak. For this reason, only the GYR data were analyzed considering separately the three axis components to minimize the impact of involuntary motion artifacts. The supine posture showed the best feasibility (98%) for this kind of analysis, with the head movements along the GYR x-axis component strongly related to the respiratory activity, thus providing a lower estimation error compared to the *y*- and *z*-axis, while breathing at 7, 10.5 and 14 breaths/ min.

In the designed protocol, the goal was to keep the observation window, during which the subject was asked to stay still, as short as possible, to mimic possible real scenarios in which the user could be guided by the VR experience to avoid or minimize head motion for the time needed for opportunistic measurements of mean HR and RESP. Three different positions (sitting, standing and supine) were investigated: the standing position, corresponding to a context of VR-gaming, or physical rehabilitation, where the subject is free to move within a confined area and interact with the VR-environment in a more realistic way; the sitting position, corresponding to a general use scenario, applicable both for gaming and healthcare applications; the supine position, for healthcare applications in the context of bedridden patients.

Based on the obtained results, the proposed approach could provide reasonably accurate results relevant to mean HR only when the subject is in sitting position and complies with the condition of not moving the head for about 20 to 30 s. Hence the users hardly keep their head still during normal VR activities, short time intervals where the subject’s head is induced to stay still could be induced by careful design of the VR application, thus allowing measuring physiological parameters, such as mean HR and RESP, that could be used to implicitly monitor the QoE, in relation to the effectiveness of the given stimuli embedded into the VR application.

As HR and respiratory activity are largely under the control of the autonomic nervous system, their measurement during a VR experience may serve as an objective index of the status of the subject in terms of vagal or sympathetic activation [31] as a surrogate measure of stress, fatigue, or emotional entrainment of the user. The possibility to achieve this aim using the inertial sensors in the VR headset, instead than using additional laboratory instrumentation or wearables, could allow ubiquitous recordings in all potential users, with no additional constraints that could affect the sympatho-vagal balance.

In particular, the HR estimation was computed by implementing similar pre-processing steps as described in [23], where cardiac information were detected from peripheral locations such as the head. By fusing the signals originating from the 3-axes ACC and those from the 3-axes GYR, the same weight is provided to each of the components, obtaining estimations more robust to different device orientations. In literature, when the location of measure is the thorax, the combination of accelerometer and gyroscope signals has been shown able to reduce noise level and overcome the limitations of using only one sensor [32,33]. However, when the location of measure is the head, the best results have been reported associated to the use of only the gyroscopic signal [23,24]. Accordingly, our choice was to assess the feasibility of both sensors without combining their output.

For the extraction of mean HR and RESP, three frequency domain approaches (FFT, STFT and Adjusted-FFT) were applied, to minimize the problem of nonlinear and nonstationary behavior in the signals, as well as the problem of missing peaks, especially for cardiac activity, due to non-constant sampling rate of the inertial measurements units. In fact, alternative approaches based on time domain analysis [26,27], mainly focused on detecting local maxima or local minima using a moving window on the acquired signals, could be more easily affected by superimposed noise, thus resulting in missing or wrong detections. The FFT method provides the power frequency information over the entire signal acquisition duration; it is less robust in presence of not perfect stationary conditions, as the normal heart rate variability. In an attempt to cope with this limitation, the STFT was also tested, thus providing a time-localized frequency analysis within a moving and overlapping 10 s time window but with lower frequency resolution that leads to discrete values of HR to be estimated. The third method (Adjusted FFT) was introduced based on a posteriori analysis of the PSD obtained using the FFT method, as a possible automated solution to the problem of finding spurious peaks in the PSD corresponding to harmonic frequencies with a power greater than the fundamental one.

To compare the obtained performance with literature, we found only one study that considered a similar experimental set-up [23], where twelve participants were studied while standing up, sitting down and lying down, under both relaxed and aroused (after biking) conditions for a minute each, while wearing Google Glasses. By dividing the 72 one minute segments into intervals of 20 s, 648 samples were collected from the participants, from which HR and the respiratory frequency were estimated separately from ACC and GYR data and compared to a gold standard. The best performances, reported pulling together sitting, standing and supine, were obtained from GYR, with 0.82 bpm bias and 7 bpm 95% LoA for HR, and 1.39 breaths/min bias and 9 breath/min 95% LoA for respiratory frequency, respectively. However, it is worth noticing that the reported LoA did not take into account for repeated measures in their Bland–Altman calculations, thus underestimating the true LoA [34]. In addition, $R^{2}$ of the linear regression were not reported; also, the sensors’ position and weight of the device (i.e., VR vs. Google Glasses) was also different. Considering each posture separately, only the mean absolute error was reported, so a real comparison cannot be performed. Our results in term of bias are comparable with [23] for SIT and STAND, where the mean absolute error was 1.18 bpm and 0.85 bpm, respectively, while they appear worst in SUP (0.44 bpm). However, considering the LoA, our results for mean HR in SUP appear better than what achieved in that study.

### Limitations

A possible limitation of our study is the need to maintain a still condition while the acquisitions are performed. As the subject is moving, the degree of the movement-related noise would exceed the physiologic components of interest; this limitation could be overcome by designing and including in the VR-experience short periods where the subject head is induced to stay still.

As the subjects were studied according to a non-randomized sequence of the analyzed postures, this acquisition protocol might have introduced a systematic error in the results obtained for each posture.

The proposed methods of analysis did not directly address the problem of determining the signal quality before processing, since this would have required the generation of a reference signal, i.e., by averaging a high number of available segments. In addition, the non-uniform sampling rate in the measurements could have introduced additional artifacts when estimating physiological information after interpolation and digital filtering, as these operations can amplify the noise [35] and negatively affect the frequency content. In addition, the frequency domain analysis does not provide information about HRV, as the signal is not analyzed in terms of beat-to-beat durations, thus limiting the capability in assessing stress response during the VR-experience: This could be explored in future dedicated studies, where possible availability of ECG devices Bluetooth connected to the VR could allow better synchronization to this aim. Additionally, as the adopted frequency domain-based methods imply stationarity of the data in the time window considered for analysis window (30 s maximum for the FFT and Adjusted-FFT, 10 s for the Short-time FFT), a transient change in the HR would not be properly captured.

Finally, a reference measure for the breathing activity was not utilized, but an experimenter was visually confirming the adherence of the subject to the controlled respiration protocol during the experimental session; in addition, from the obtained results, it was evident that the imposed frequencies were properly followed by each subject.

## 5. Conclusions

In this study, the feasibility of deriving cardio-respiratory parameters using a VR head-worn device, exploiting the subtle beat-to-beat head motion associated with the mechanical activity of the heart, was explored. Although not designed to acquire physiological signals, the results provided evidence of the feasibility of the proposed approach, with a degree of performance and feasibility dependent on the posture of the subject, under the conditions of keeping the head still for a short period (20-30 s). The gyroscope outperformed the accelerometer, demonstrating that rotational movements of the head are more related to cardio-respiratory activity. In particular, the best agreement with ECG gold standard in HR was found while sitting; conversely, the supine posture showed the best feasibility to allow the estimation of the respiratory rate from each gyroscope axis, leading to the identification of the transversal direction as the one containing higher breathing information The proposed approach has the potential to allow for opportunistic monitoring of mean HR and respiration to provide, in an implicit way, a bio feedback of the user’s QoE to multimedia contents or VR scenarios.

## Figures and Tables

**Figure 1 sensors-20-07168-f001:**
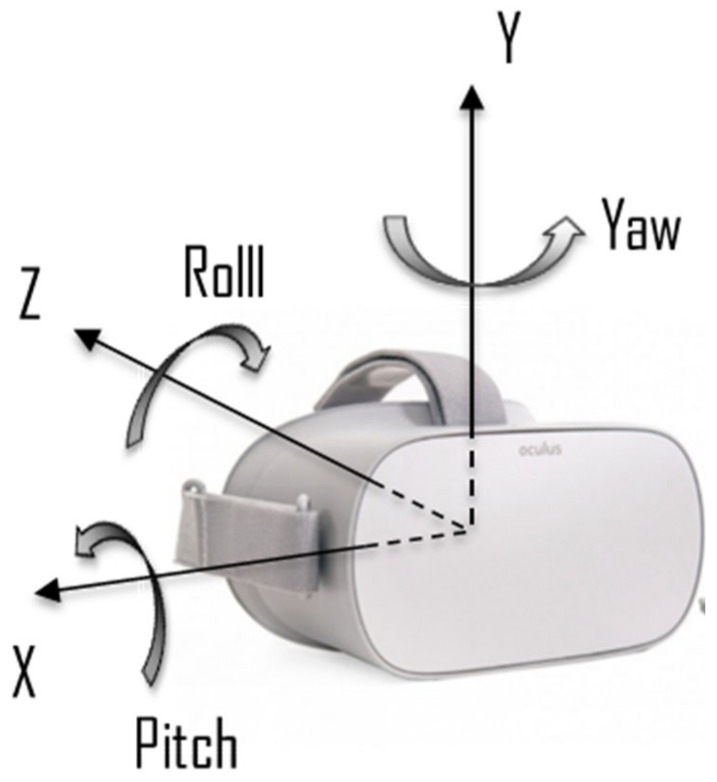
The VR device used for the experiments, with its reference for linear and rotational accelerations indicated.

**Figure 2 sensors-20-07168-f002:**
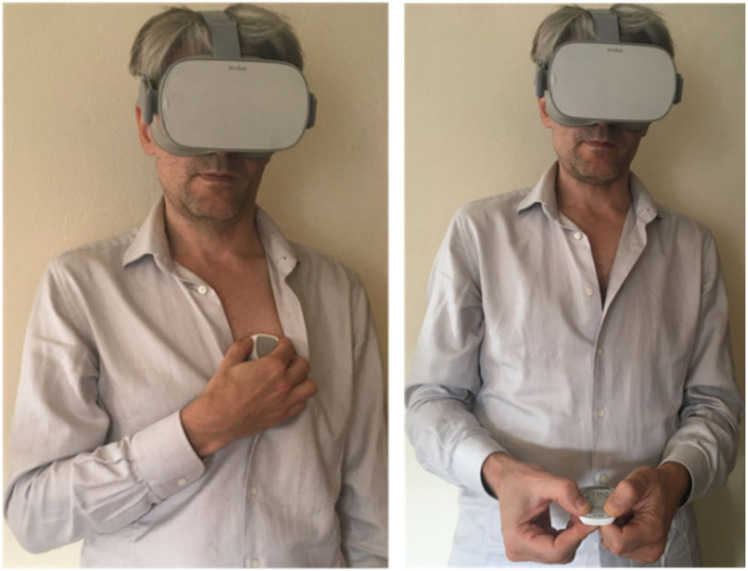
In the experimental protocol, Coala^®^ Heart rate Monitor device [30] was used to obtain gold standard ECG measurement during VR signals acquisition: first, it is placed on the chest for the first measurement (ECG-Chest), and then, it is held between the thumb and the middle fingers for the second measurement (ECG-Thumb).

**Figure 3 sensors-20-07168-f003:**
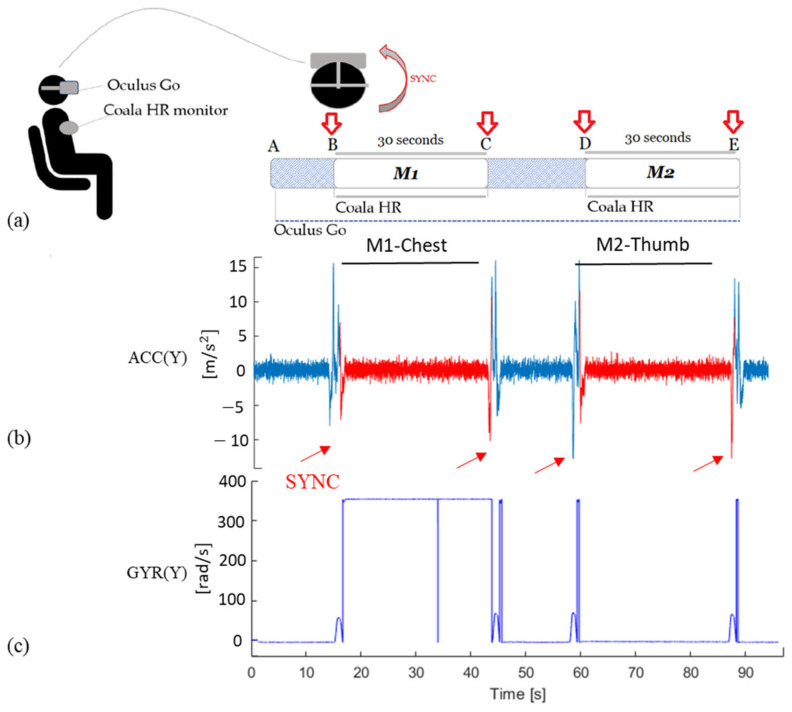
Schematic representation of HR acquisition protocol (**a**). The red arrows (SYNC) indicate the artifacts induced by fast head rotation on the VR acquired signals, in correspondence to each start and end of the ECG recordings; this resulted in a clear artifact visible on the y-axis of the ACC signal (**b**) thus allowing the automatic extraction of the portion of ACC and GYR signals (M1-Chest and M2-Thumb) corresponding in time with the gold standard ECG-measurements (ECG-Chest and ECG-Thumb). In (**c**) the y-axis of the GYR signal corresponding to the same subject in SIT is shown, highlighting how the SYNC artifacts could not be univocally identifiable from it.

**Figure 4 sensors-20-07168-f004:**
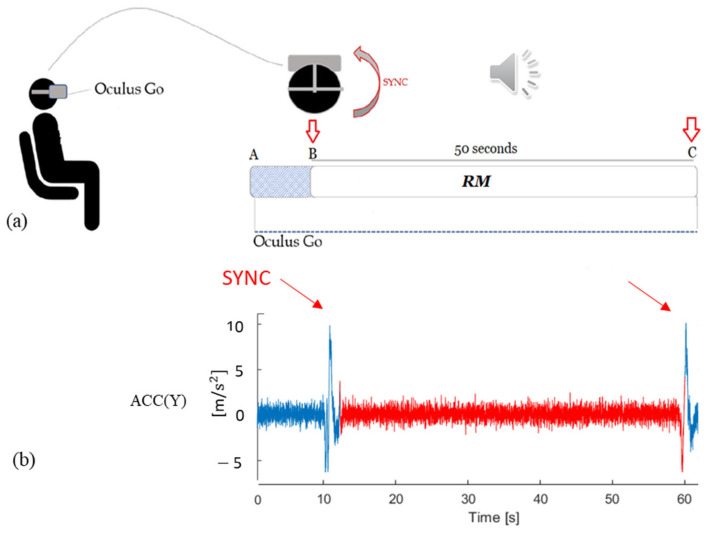
Schematic representation of the protocol for RESP evaluation (**a**). The red arrows (SYNC) indicate the artifacts induced by fast head rotation on the VR acquired signals in correspondence to each start and end of the imposed breathing through audio listening, clearly visible on the y-axis of the ACC signal (**b**), thus allowing the automatic extraction of the portion of ACC and GYR signals acquired under controlled breathing (RM).

**Figure 6 sensors-20-07168-f006:**
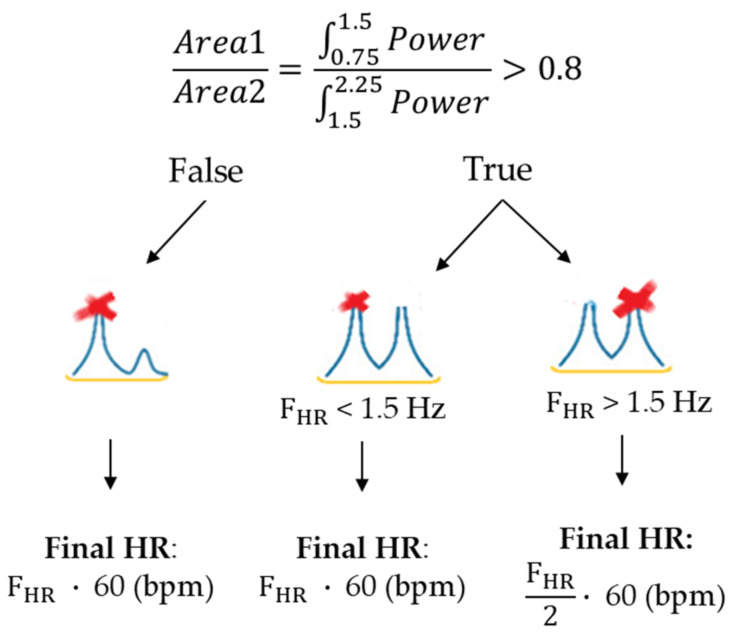
A schematic representation of the steps in the Adjusted-FFT Method: if the power ratio was >0.8, an additional test was performed to detect the proper frequency corresponding to HR.

**Figure 7 sensors-20-07168-f007:**
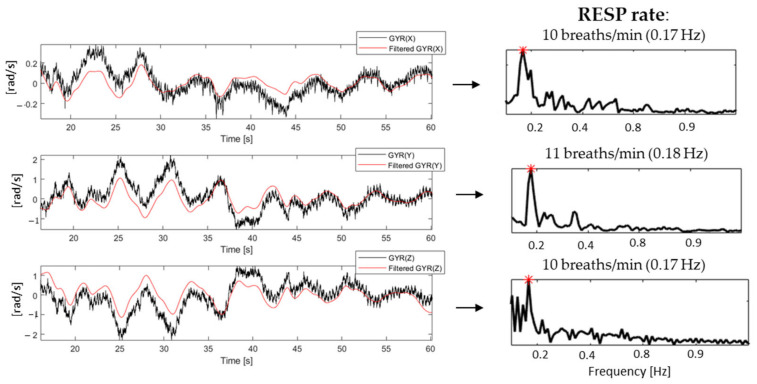
On the left, an example of the three GYR components (in black) with the corresponding output of the applied filter superimposed (in red), and on the right, the corresponding PSD with the frequency with maximum power estimated as RESP rate highlighted.

**Figure 8 sensors-20-07168-f008:**
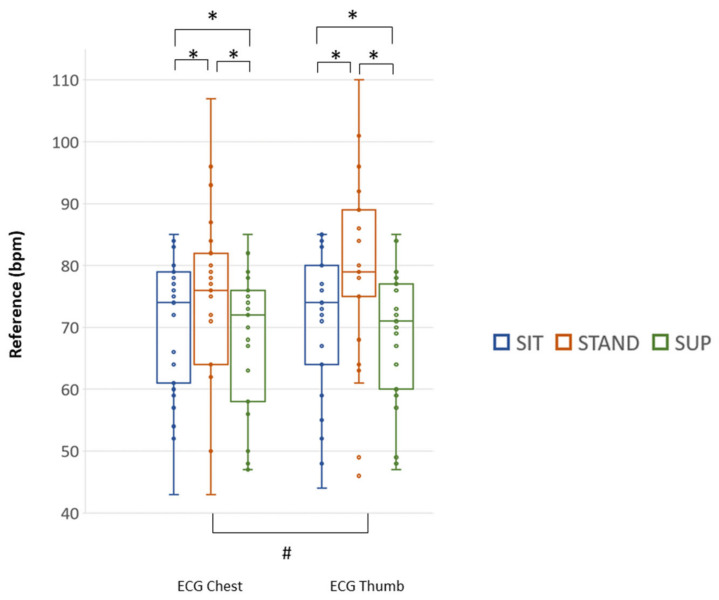
The median HR (25th; 75th percentiles), expressed in bpm, and individual data (as dots) measured from the chest and the thumb with the reference Coala Heart monitor in the two consecutive acquisitions (M1-Chest and M2-Thumb) for each body posture in SIT (blue), STAND (red) and SUP (green) in the 25 subjects that allowed paired comparisons for all conditions. # *p* < 0.01 ECG Chest vs. ECG Thumb (Wilcoxon signed rank test); * *p* < 0.05 between postures (Friedman Test and Wilcoxon signed rank test).

**Figure 9 sensors-20-07168-f009:**
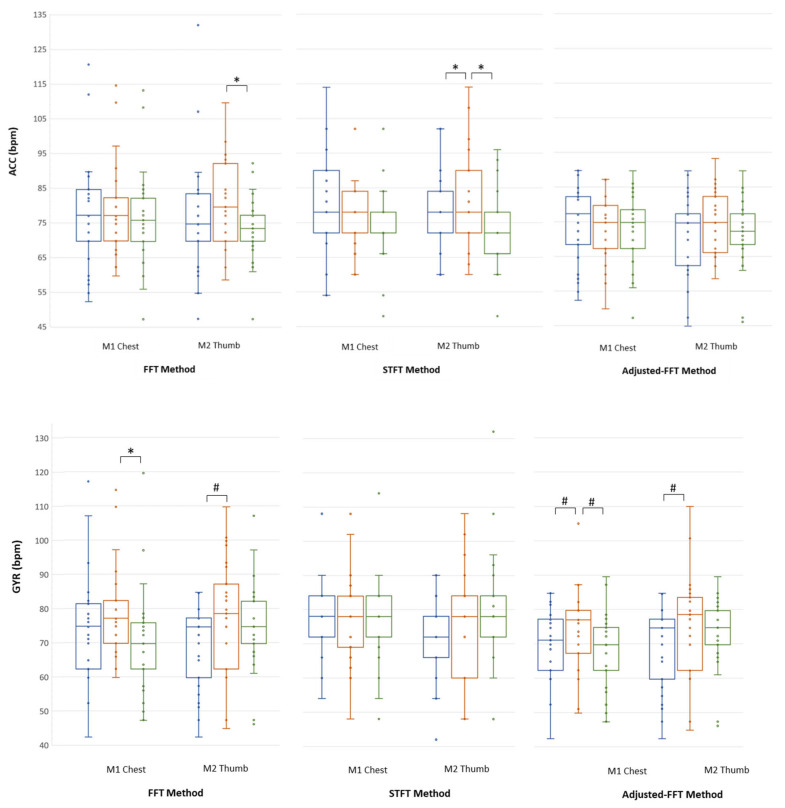
The median ${\mathrm{HR}}_{\mathrm{Oculus}}$ (25th; 75th percentiles) and individual data (as dots), expressed in bpm, measured from ACC (top panel) and GYR (bottom panel) signals obtained by M1-Chest and M2-Thumb, separately, for each method and body posture in SIT (blue), STAND (red) and SUP (green) in the 25 subjects that allowed paired comparisons in all conditions. * *p* < 0.05 and # *p* < 0.01 between postures (Friedman Test and Wilcoxon signed rank test).

**Figure 10 sensors-20-07168-f010:**
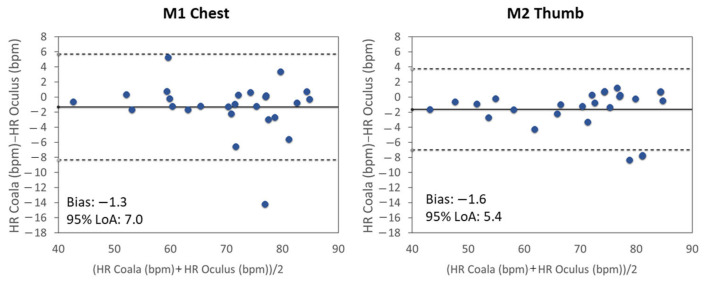
Bland–Altman analyses between the gold standard $\left({\mathrm{HR}}_{\mathrm{Coala}}\right)$ and the mean HR measurements obtained from the GYR signal $\left({\mathrm{HR}}_{\mathrm{Oculus}}\right)$ of the M1-Chest (left) and M2-Thumb (right) acquisitions while in SIT position using the Adjusted-FFT method.

**Table 1 sensors-20-07168-t001:** Results of Spearman rank-order correlation (R_s_^2^) reported together with lower and upper limits of the respective 95% confidence interval (lower-upper) relevant to HR estimated from the ACC and GYR signals compared to the gold standard, by applying the three different methods tested, in SIT, STAND and SUP, separately for the first (M1 Chest) and s (M2 Thumb) acquisitions. The number of subjects (N) included in each comparison is included. The best performance for each posture is highlighted in bold.

			ACC			GYR	
Method\Posture		SIT (N = 26) R_s_^2^	STAND (N = 28) R_s_^2^	SUP (N = 29) R_s_^2^	SIT (N = 26) R_s_^2^	STAND (N = 28) R_s_^2^	SUP (N = 29) R_s_^2^
**FFT**	**M1** **Chest**	0.28 (0.03–0.58)	0.23 (0.02–0.52)	0.00 (0.14–0.33)	0.58 (0.29–0.79)	0.79 (0.60–0.90)	0.30 (0.05–0.58)
	**M2** **Thumb**	0.17 (0–0.47)	0.08 (0.01–0.35)	0.06 (0.02–0.31)	0.91 (0.81–0.96)	0.81 (0.64–0.91)	0.14 (0–0.43)
**STFT**	**M1** **Chest**	0.23 (0.01–0.54)	0.31 (0.06–0.59)	0.1 (0.09–0.18)	0.23 (0.01–0.54)	0.76 (0.55–0.88)	0.1 (0–0.38)
	**M2** **Thumb**	0.2 (0.07–0.25)	0.15 (0.1–0.18)	0.1 (0–0.38)	0.49 (0.19–0.73)	0.79 (0.60–0.90)	0.04 (0.03–0.27)
**Adjusted** **FFT**	**M1** **Chest**	0.44 (0.14–0.7)	0.12 (0–0.41)	0.1 (0.05–0.24)	0.83 (0.66–0.92)	0.51 (0.21–0.73)	0.44 (0.15–0.68)
	**M2** **Thumb**	0.55 (0.25–0.77)	0.11 (0–0.4)	0.02 (0.01–0.49)	0.91 (0.81–0.96)	0.69 (0.43–0.84)	0.20 (0.01–0.48)

**Table 2 sensors-20-07168-t002:** Results of Bland–Altman analysis reported as Bias (95% LoA) relevant to HR (in bpm) estimated from the ACC and GYR signals compared to the gold standard, by applying the three different methods tested, in SIT, STAND and SUP, separately, for the first (M1 Chest) and second (M2 Thumb) acquisitions. The standardized effect size ($\frac{\Delta M}{SD}$ ) as the difference between the mean of the two groups divided by the standard deviation (SD), is also reported. The number of subjects (N) included in each comparison is included. The best performance for each posture is highlighted in bold for GYR signal.

				ACC							GYR			
		SIT (N = 26)		STAND (N = 28)		SUP (N = 29)			SIT (N = 26)		STAND (N = 28)		SUP (N = 29)	
		Bias (LoA)	$\frac{\Delta M}{SD}$	Bias (LoA)	$\frac{\Delta M}{SD}$		Bias (LoA)	$\frac{\Delta M}{SD}$	Bias (LoA)	$\frac{\Delta M}{SD}$	Bias (LoA)	$\frac{\Delta M}{SD}$	Bias (LoA)	$\frac{\Delta M}{SD}$
**FFT**	**M1** **Chest**	6.9 (31.2)	0.249	4.7 (30.3)	0.240		11 (39)	0.374	3.6 (25.3)	0.131	0.6 (14.9)	0.049	6.6 (38.9)	0.321
	**M2** **Thumb**	5.3 (37.2)	0.178	5.5 (42.9)	0.254		5.8 (27.6)	0.361	1.9 (25.4)	0.068	−1.7 (11.6)	0.046	5.8 (29)	0.203
**STFT**	**M1** **Chest**	5.7 (29.4)	0.229	4.6 (27.2)	0.242		10.1 (34.3)	0.521	5.7 (29.4)	0.229	0.7 (15.1)	0.034	10.6* (34.1)	0.358
	**M2** **Thumb**	5.5 (33)	0.236	6.4 (42.6)	0.199		7.1 (26.3)	0.371	1.5 (19.5)	0.064	−2.9 (11.7)	0.097	11 (37.8)	0.326
**Adjusted** **FFT**	**M1** **Chest**	3.4 (19.2)	0.154	−0.8 (28)	0.002		5.6 (27.3)	0.351	−1.3 (7)	0.060	−1.3 (22.6)	0.037	0.2 (19.4)	0.124
	**M2** **Thumb**	0.2 (13.7)	0.001	−2.4 (36.9)	0.026		4.3 (23)	0.273	−1.6 (5.4)	0.068	−3.6 (13.1)	0.127	4.3 (21)	0.146

**Table 3 sensors-20-07168-t003:** Number of acquisitions that were discarded for each specific body position (SUP, SIT and STAND) by considering the respiratory signal derived from the *x*-, *y*- and *z*-axis of the GYR for each imposed RESP rate, set at 7, 10.5 and 14 breaths/min.

	SUP			SIT			STAND		
**Reference (breaths/min)**	**X**	**Y**	**Z**	**X**	**Y**	**Z**	**X**	**Y**	**Z**
7	0	0	0	5	4	3	8	7	8
10.5	0	1	1	3	4	5	7	7	11
14	0	0	0	4	2	5	5	4	6
**Total Number** **of discarded acquisition**	2			35			63		

**Table 4 sensors-20-07168-t004:** The median (25th; 75th percentiles), expressed in breaths per minute, of the RESP rate distribution obtained for each imposed RESP rate in the three body postures for x-y and z-axis.

	SUP (N = 88)			SIT (N = 55)			STAND (N = 27)		
**Reference** **(breaths/min)**	**X**	**Y**	**Z**	**X**	**Y**	**Z**	**X**	**Y**	**Z**
**7**	7 (7; 7)	7 (7; 10)	7 (6; 9)	7 (7; 7)	7 (6; 9)	10 (7; 12)	7 (7; 9)	7 (5; 8)	7 (7; 12)
**10.5**	10 (10; 11)	10 (9; 11)	10 (9; 11)	10 (10; 11)	10 (9; 11)	10 (10; 11)	10 (10; 11)	10 (1; 10)	10 (10; 13)
**14**	14 (14; 14)	11 (9; 14)	11 (6; 14)	14 (14; 14)	14 (9; 14)	14 (8; 14)	14 (14; 15)	12 (7; 14)	14 (10; 14)

**Table 5 sensors-20-07168-t005:** The median absolute error, expressed in breaths per minutes, of the RESP frequency is computed from the *x*-, *y*- ad *z*-axis GYR for each posture. The median (25th; 75th) is reported considering the three different imposed respiratory rates.

	SUP (N = 88)			SIT (N = 55)			STAND (N = 27)		
**Reference** **(breaths/min)**	**X**	**Y**	**Z**	**X**	**Y**	**Z**	**X**	**Y**	**Z**
**7**	0.5 (0.5; 0.8)	1.7 (0.5; 6.7)	1.7 (0.5; 3.3)	0.5 (0.5; 1.7)	1.1 (0.5; 3.7)	4.2 (0.8; 5.6)	0.6 (0.5; 2.5)	1.4 (0.5; 3.2)	0.8 (0.5; 5.1)
**10.5**	0.6 (0.5; 0.7)	0.7 (0.5; 3.0)	1.8 (0.7; 4.3)	0.5 (0.5; 1.8)	1.8 (0.5; 4.3)	0.7 (0.5; 2.0)	0.7 (0.5; 1.9)	0.7 (0.5; 4.9)	0.7 (0.5; 3.7)
**14**	0.3 (0.3; 0.9)	2.8 (0.3; 5.3)	2.8 (0.3; 7.8)	0.3 (0.3; 0.3)	1.3 (0.3; 5.3)	1.2 (0.3; 5.6)	0.3 (0.3; 1.6)	1.3 (0.3; 6.4)	1.5 (0.3; 4.4)

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
