# Peer review of "Feasibility of Heart Rate and Respiratory Rate Estimation by Inertial Sensors Embedded in a Virtual Reality Headset"

_sensors, 2020, doi:10.3390/s20247168_

Round 1
Reviewer 1 Report
The paper was well written. The reviewer provided some issues to increase the quality of the method.
Abstract:
The CI 95% must be added to the R-square values.
Line 91,
The link of the “Polytechnic University of Milan (date of approval: 10 September 2019)” in the paper is the following hyperlink:
https://en.wikipedia.org/wiki/Polytechnic_University_of_Milan
It is not clear why it was used. Please provide the ethical code of the approved project.
Line 106
The ACC and GYR signals were recorded in the experiment. Both of them have noise, and they are usually combined using different methods. The reviewer highly recommends trying such algorithms to see whether the performance of the method increases.
Moreover, linear regression was used in this study. It is recommended to use the robust linear regression to see if the performance increases: [chapter 8, robust regression]
https://epubs.siam.org/doi/book/10.1137/1.9781611973655?mobileUi=0
It might be interesting to add the demographic variables to the MLR method to see if it improves the performance.
The performance of the proposed method is not high. Thus, It is recommended to use advanced methods such as LSTM or other time-sample prediction models to use the fusion of the ACC and GYR data to estimate the instantaneous HR. Using this method, it will be possible also to estimate HR variability.
Fig.3, please correct section (a) as two items were overlapped.
Line 147, Please also cite any MATLAB toolbox used in this study.
Line 184, Were the analyzed epochs stationary to estimate the FFT?
In tables 1 and 2, the CI 95% must be added to R-square.
Figure 10: It is excellent that the authors used the Bland Altman plot. Based on the plot, how do the authors interpret the problems of the proposed method?
Discussion:
The comparison with the state-of-the-art is missing.
Reviewer 2 Report
The Figures 3-6 need the axis values for the Y axis shown. The Figure 5 is not an informational Picture, and could be corrected.
Author Response
Reply to Reviewer 2
We would like to thank the Reviewer for his/her comments, that we believe helped in increasing the quality of our manuscript. Here below the replies in a point-by point basis:
- The Figures 3-6 need the axis values for the Y axis shown.
We thank the Reviewer for noticing it. As requested, the axis values are now shown.
- The Figure 5 is not an informational Picture, and could be corrected.
As requested, Fig.5 has been removed.
Reviewer 3 Report
Introduction
1. L34: The acronym is incorrect. It should be “Virtual and augmented reality (VAR)…” or “Virtual reality (VR) and augmented reality (AR)…” I’m not in favour of using acronyms though when it is just as easy to use the full term.
2. L44: What is the definition of ‘quality of experience’? Later you describe how it is measured, but you first need to describe what it is.
3. L47-50: It’s not clear to me why physiological data would represent the ‘quality’ of experience. You first need to define what is meant by ‘quality’ (comment #2), but if ‘quality’ refers to ‘how good it was’ (i.e. better or worse) or something similar, then I’m not sure why a higher heart rate would necessarily mean a ‘better’ experience. I think you need to provide a better rationale for the link between quality of experience and physiological data.
4. L61-64: The study by Hernandez et al (2015) failed to account for repeated measures in their Bland-Altman calculations. Despite having a small sample of 12 participants, the Bland-Altman LoA were calculated on 648 pairs of data. Yet according to Bland and Altman (2007) this will produce LoA that are too narrow. The LoA reported are about ± 7 beats/min, which will be an underestimation of the true LoA. You need to acknowledge this.
Bland, J. M., & Altman, D. G. (2007). Agreement Between Methods of Measurement with Multiple Observations Per Individual. Journal of Biopharmaceutical Statistics, 17(4), 571–582. https://doi.org/10.1080/10543400701329422
5. The Introduction is overly positive in my view. I think you need to discuss some of the potential sources of error when using ballistocardiographic measures. For example, virtual reality games (and other uses) often involve substantial body movements. How might movement artefact affect the accuracy of heart rate and respiratory rate measures?
Method
6. L88: Why 30? Did you perform an a priori sample size estimation?
7. L88: ‘height’ should be ‘stature’.
8. L89: ‘weight’ should be ‘body mass’.
9. L89: ‘Kg’ should be ‘kg’.
10. L96: How was the fitting of the VR headset standardised? Was a certain level of tightness applied equally across participants? This is important because you are assuming that the head and VR headset are a fixed rigid structure.
11. L106-107: Please provide more information on the SDK used. Which version of the SDK and which APIs were used? Is the application source code available for examination?
12. L113-120: How soon after sitting, standing, and lying down were the heart rate measurements collected?
13. L128: Was the order randomised?
14. L136: Did you measure respiratory rate directly, using indirect calorimetry?
15. L213: Did you use both heart rate means for each condition? That is, both M1 and M2 in Figure 3? If so, did you account for repeated measurements as per Bland and Altman (2007) mentioned above?
16. L216 and L218: Did you test the data for normality? If so, how?
Results
17. L230-232: I’m not sure about the rationale for discarding data like this. By doing this you are removing a source of error from the analysis and therefore improving the agreement between devices.
18. L241: Please report the mean difference and 95% confidence interval for the mean difference for all pairwise comparisons. You should also report standardised effect sizes for all pairwise comparisons.
19. Figure 9: Please provide the individual data points on these bar plots.
Weissgerber, T. L., Milic, N. M., Winham, S. J., & Garovic, V. D. (2015). Beyond bar and line graphs: time for a new data presentation paradigm. PLoS Biology, 13(4), e1002128.
20. Tables 1 and 2: I’m assuming ‘CI’ is the limits of agreement? If so, please change to ’95% LoA’.
21. Figure 10: You say on L88 that 30 participants were recruited. Then on L236/7 you revise that figure depending on condition to 26, 28, and 29. Yet in Figure 10 the Bland-Altman (and correlation) plot shows upwards of 40 data points. See comment #14 in relation to using repeated measures in a Bland-Altman limits of agreement calculation.
Discussion
22. Not until L341 are the results of the study discussed. From L282 to L340 is all mostly a rehash of the method. As such, please start the discussion with the main findings.
23. L291: ‘respirator’ should be ‘respiratory’.
24. L291-297: Yet in the introduction your rationale was based on ‘quality of experience’. There is inconsistency here that you should address.
25. L353: You haven’t discussed the actual results though. How well do the VR measures agree with the ECG measures? That is, please discuss the LoA data and how those limits of agreement could be used in the field.
26. L353-354: “provided the subject complies with the conditions of not moving the head for about 30 seconds”. This is a key point that needs further discussion. How often would this occur in reality when using a VR system?
27. L379-383: Alternatively, the user could just wear a heart rate monitor (e.g. Polar) and obtain ECG-accurate data. I guess my point is that maybe the VR solution is analogous to using a sledge hammer to crack a walnut.
28. L409-410: “The proposed approach offers the possibility to capture user perception during a VR experience”. No, I don’t think it does. Heart rate and respiratory rate are not measures of perception.
Round 2
Reviewer 1 Report
The paper is suitable for publication.Reviewer 3 Report
The authors have answered all of my queries. Thank you.